# Evaluation of Vitamin D isolated or Associated with Teriparatide in Peri-Implant Bone Repair in Tibia of Orchiectomized Rats

**DOI:** 10.3390/biology12020228

**Published:** 2023-01-31

**Authors:** Pedro Henrique Silva Gomes-Ferreira, Paula Buzo Frigério, Juliana de Moura, Nathália Dantas Duarte, Danila de Oliveira, Joseph Deering, Kathryn Grandfield, Roberta Okamoto

**Affiliations:** 1Department of Diagnosis and Surgery, Araçatuba Dental School, São Paulo State University Júlio de Mesquita Filho—UNESP, Araçatuba 16018-805, Brazil; 2Department of Basic Sciences, Araçatuba Dental School, São Paulo State University Júlio de Mesquita Filho—UNESP, Araçatuba 16018-805, Brazil; 3Department of Materials Science and Engineering, McMaster University, Hamilton, ON L8S 4L8, Canada; 4School of Biomedical Engineering, McMaster University, Hamilton, ON L8S 4L8, Canada

**Keywords:** osteoporosis, vitamin D, teriparatide, men, dental implants

## Abstract

**Simple Summary:**

The teriparatide is a drug used to treatment of severe osteoporosis, it is a tissue anabolic agent, this drug increases bone mineral density and increased bone mass. Vitamin D is important to balance of serum calcium and increased calcium absorption from intestine, therefore can be applied to optimize the treatment of osteoporosis. The present study aimed to evaluate the peri-implant bone tissue in rats submitted to orchiectomy, treated with vitamin D alone or associated with teriparatide. Through this in vivo study, it was possible to conclude that the treatment with vitamin D associated with teriparatide in rats with induced osteoporosis increases the volume and improves the bone quality around the implants installed. However, the verification of the behavior of medications that promote an increase in bone density, enabling the establishment of a clinical protocol for the treatment of osteoporosis, is essential, in view of the increase in the population with this bone metabolism disorder in dental offices looking for rehabilitation treatment with dental implants.

**Abstract:**

(1) Background: The objective of this study was to evaluate the morphometry of peri-implant bone tissue in orchiectomized rats, treated with vitamin D isolated or associated with teriparatide. (2) Methods: 24 rats were divided into 4 groups: ORQ—orchiectomy, without drug treatment, ORQ+D—orchiectomy, treated with vitamin D, ORQTERI—orchiectomy, treated with teriparatide and ORQTERI+D—orchiectomy, treated with teriparatide + vitamin D. Each animal received an implant in the tibial metaphysis. Euthanasia occurred 60 days after implant surgery. Computed microtomography (micro-CT) was performed to evaluate the parameters of volume and percentage of bone volume (BV, BV/TV), trabecular thickness (Tb.Th), number and separation of trabeculae (Tb.N, Tb.Sp) and percentage of total porosity (Po-tot). Data were subjected to 1-way ANOVA and Tukey post-test, with a significance level of 5%. (3) Results: For the parameters BV, BV/TV, Tb.Th, the ORQTERI+D group showed the highest values in relation to the other groups and for Po-tot, the lowest values were for ORQTERI+D. For Tb.Sp and Tb.N, there was no statistically significant difference when comparing intragroup results (*p* > 0.05). (4) Conclusions: It was possible to conclude that treatment with vitamin D associated with teriparatide increases bone volume and improves bone quality.

## 1. Introduction

Quality of bone tissue is a determining factor for the success of osseointegration in implant-supported rehabilitations, considering mainly intrinsic characteristics of bone tissue such as structural remodeling (resorption/neoformation) [1,2,3,4]. When resorption surpasses neoformation, a lower bone density is manifested, compromising the bone/implant interface [1] and characterizing the condition of osteoporosis, observed mainly in postmenopausal women [5]. However, it is also reported in men, with a higher incidence after the 6th decade of life [5,6].

The diagnosis of osteoporosis is classified by the World Health Organization (WHO) according to a bone mineral density (BMD) test using the T-score (Normal: −1.0 or greater; low bone mass (osteopenia): between −1, 0 and −2.5, osteoporosis: < or =−2.5, severe osteoporosis: < or =−2.5 associated with fragility fracture) [7]. For this, the BMD of the lumbar spine (L1L4), hip, and femoral neck is evaluated [8]. The National Health Bone Alliance in the USA [9], recommends that postmenopausal women and men over the age of 50 should be diagnosed with osteoporosis in the following situations: T-score ≤ −2.5 in the spine lumbar or hip; low-intensity trauma resulting in hip fracture; low BMD osteopenia with vertebral fracture, humerus, pelvis, or in some cases, forearm.

Among the causes of osteopenia in males, in addition to advanced age, hypogonadism may also result in a high rate of remodeling and acceleration of bone loss [10,11,12]. The aromatization of testosterone to estradiol plays an important role in regulating bone homeostasis in men, limiting age-related bone loss [6,11,13,14,15,16,17,18]. Other factors described in the literature include vitamin D insufficiency, decreased calcium absorption, poor diet or even senility [11,18,19,20].

Vitamin D derived from enriched dairy products and fish oils or by ultraviolet irradiation [21,22], has its endogenous active form 1,25(OH)2D3 (synthesized by the kidneys), acting on calcium homeostasis and increasing calcium absorption from the intestine. In senile patients, the combined effect of decreased intestinal calcium absorption and a decline in the kidney’s ability to synthesize 1,25(OH)2D3 may be the cause of secondary hyperparathyroidism, increasing bone catabolism to achieve homeostasis of serum calcium [23]. Therefore, vitamin D supplementation seems to be one of the alternatives widely used when considering the maintenance of serum calcium homeostasis, available in skeletal health, and can be applied to optimize the treatment of osteoporosis [24,25,26,27].

Some drug substances are indicated for the control of less severe osteoporosis, such as agents that promote bone resorption (denosumab, raloxifene or bisphosphonates) [6]. However, for more severe osteoporosis, drugs such as teripartide (Forteo, Eli Lilly, Indianapolis, IN, USA), also called rhPTH (1–34), an analogue of parathyroid hormone, is one of the most indicated, as it is a tissue anabolic agent. Teriparatide increases bone mineral density directly by activating osteoblasts, and indirectly by increasing renal tubular reabsorption of calcium and its intestinal absorption, stimulating new bone formation and increased bone mass [28,29,30,31]. When associated with vitamin D, it presents with greater bone formation through the increase of osteoblastic function [32,33,34]. Teriparatide is an effective option and is currently the only bone anabolic substance approved for the treatment of male osteoporosis [30,31].

However, the verification of the behavior of medications that promote an increase in bone density, enabling the establishment of a clinical protocol for the treatment of osteoporosis, is essential, in view of the increase in the population with this bone metabolism disorder in dental offices. seeking rehabilitation treatment with dental implants. Therefore, the present study aimed to evaluate the morphometry of peri-implant bone tissue in rats submitted to orchiectomy, treated with vitamin D alone or associated with teriparatide.

## 2. Materials and Methods

### 2.1. Animals

The sample number of the present study for each group was determined using the power test through OpenEpi (Version 3, open-source calculator), based on previous results already published by Gomes-Ferreira et al. 2020 [35]: the means used for the calculation were 3.06 and 4.898 and the standard deviations were 0.26 and 0.024, with a significance level of 5% and power of 95% in a one-tailed hypothesis test.

As approved by the Research Ethics Committee on the use of animals at the Faculty of Dentistry of Araçatuba—UNESP (FOA Process n° 2015-00238), 24 male rats (Rattus norgicus albinus, Wistar) weighing approximately 500 g were used in this study. study, divided into four experimental groups (ORQ, ORQ+D, ORQTERI and ORQTERI+D) (Table 1), according to the osteoporosis induction surgery and drug treatment to which they were submitted.

### 2.2. Bilateral Orchiectomy

For induction of osteoporosis, 24 rats in ORQ, ORQ+D, ORQTERI and ORQTERI+D groups were anesthetized with Coopazine (Xilazine-Coopers, Brazil, Ltd., Osasco, SP, Brazil) and Vetaset (Injectable Ketamine Hydrochloride, Fort Dodge, Health Animal Ltd., Campinas, SP, Brazil) and, afterwards, incisions were made in both scrotal sacs, exposing the testicles. With hemostatic forceps, the spermatic cord was presented, with concomitant individualization and ligation of the vas deferens and vascular pedicle, and then sectioned. The testicle was removed, and the surgical wound was sutured with 4-0 silk thread (Ethicon, Johnson & Johnson, São José dos Campos, SP, Brazil).

### 2.3. Drug Treatment

From the day of orchiectomy surgery, the ORQ, ORQ+D, ORQTERI and ORQTERI+D groups maintained a conventional diet and water ad libitum. After 30 days, drug treatment was started. The ORQ+D group was submitted to drug treatment through daily gavage with vitamin D (Borachi Vieira Ribas & Cia Ltd., Araçatuba, SP, Brazil) in the dose of 0.1 mcg/kg/day, the ORQTERI group was submitted to drug treatment through daily subcutaneous application with Teriparatide (Forteo—Eli Lilly, Indianapolis, IN, USA) (0.5 mcg/kg/day), the ORQTERI+D group was submitted to drug treatment through daily subcutaneous application with Teriparatide (Forteo—Eli Lilly, Indianapolis, IN, USA) (0.5 mcg/kg/day) [35,36,37] plus daily vitamin D gavage (0.1 mcg/kg/day) [38] and ORQ submitted to gavage and subcutaneous application of 0.9% isotonic sodium chloride solution (Physiologic^®^, Laboratories Biosynthetic Ltd., Ribeirão Preto, SP, Brazil), to maintain the same stress. For this same reason, the ORQ+D group was also submitted to a subcutaneous injection of an isotonic solution.

### 2.4. Surgery to Install Implants in Tibias of Rats

After 30 days of drug treatment, the 24 rats underwent implant placement in the tibias. Twenty-four commercially pure grade IV titanium implants were installed with a surface treated by double acid etching (nitric, hydrofluoric, and sulfuric acids), with a diameter of 1.5 mm and a height of 3.5 mm, sterilized by gamma rays. For this, milling was performed with a 1.3 mm diameter spiral milling cutter mounted on an electric motor (BLM 600^®^; Driller, São Paulo, SP, Brazil) at a speed of 1000 rpm, under irrigation with an isotonic solution of sodium chloride. at 0.9% (Physiologic^®^, Laboratories Biosynthetic Ltd., Ribeirão Preto, SP, Brazil), and contra-angle with 20:1 reduction (Angle 3624N 1:4, Head 67RIC 1:4, KaVo^®^, Kaltenbach & Voigt GmbH & Co., Biberach, Baden-Württemberg, Germany) and a depth of 3.0 mm, with locking and primary stability (Figure 1A–H).

Each animal received an implant in the tibial metaphysis. The tissues were sutured in planes using absorbable thread (Polygalactin 910—Vycril 4.0, Ethicon, Johnson Prod., São José dos Campos, SP, Brazil) with continuous stitches in the deep plane and with multifilament thread (4-0 silk, Ethicon, Johnson, São José dos Campos, Brazil) with interrupted points on the outermost plane. In the immediate postoperative period, each animal received a single intramuscular dose of 0.2 mL of Penicillin G-benzathine (Pentabiotic, Fort Dodge Health Animal Ltd., Campinas, SP, Brazil). Euthanasia of these animals was performed 60 days after implant placement.

### 2.5. Microtomographic Analysis (Micro-CT)

After euthanasia of the animals, the tibias of the 4 experimental groups (ORQ, ORQ+D, ORQTERI and ORQTERI+D) were reduced and fixed in 10% buffered formalin solution (Analytical Reagents, Dynamic Ltd., Catanduva, SP, Brazil) for 24 h and bathed in running water for 24 h. After fixation, the pieces were left in 70% alcohol for the microtomographic analysis.

Through computed micro-tomography (micro-CT) (SkyScan 1272 Bruker MicroCT, Aatselaar, Antuérpia, Belgium, 2003), the pieces were scanned using 6 µm thick slices (90 kV and 111 μA) with a 0.5 mm Al + Cu filter, 0.038° rotation step, camera size of 2016 × 1344, and acquisition time of 1 h and 32 min (Figure 2). The images obtained by projecting X-rays for the samples were stored and reconstructed, determining the area of interest using the NRecon software (SkyScan, 2011; Version 1.6.6.0), with smoothing of 1, artifact rings correction of 8, Beam correction Hardening of 24% and the image conversion ranged from 0.0–0.14. In the Data Viewer software (SkyScan, Version 1.4.4 64-bit) the images were reconstructed, and observed in three planes (transverse, longitudinal and sagittal). Then, using the software CTAnalyser—CTAn (2003-11SkyScan, 2012 BrukerMicroCT Version 1.12.4.0) the bone volume (BV), percentage of bone volume (BV/TV), trabecular bone thickness (Tb.Th) were defined, separation and number of trabeculae (TB.Sp and Tb.N) and total porosity (Po-tot) were measured. 3D reconstructions were performed using the CTvox software (SkyScan, Version 2.7). This processing sequence is presented in Figure 3.

After the scanning, the analyzes of pieces referring to ORQ, ORQ+D, ORQTERI and ORQTERI+D groups were carried out within 60 days after the implants were installed.

### 2.6. Preparation of Calcified Tissues

After micro-CT analysis, the samples were progressively dehydrated in ethanol (70%, 80%, 90% and 100%). At the end of dehydration, the pieces were immersed in a mixture of 100% alcohol and Techno Vit^®^ light-curing resin (Kulzer GmbH, Hanau, Germany) in different concentrations, until only resin remained as an immersion medium, and the Technovit resin was light-cured. Embedded blocks were then cut along the medio-distal plane using a cutting system (Exakt Cutting System, Apparatebau, Gmbh, Hamburg, Germany) and polished using 400 to 2400 grit silicon carbide paper to expose the cross-section of the bone-biomaterial interface. The samples were fixed to SEM stubs with carbon tape, wrapped in aluminum tape, and then painted with nickel when necessary. Later, they were coated with a thin layer of carbon (∼10 nm) to improve the conductivity in the SEM. The workflow of all electron microscopy characterizations at the bone-Biogran^®^ interface is described in detail by Micheletti et al., 2021 [39] but the main steps are reported herein.

### 2.7. Acid Etching and Scanning Electron Microscopy (SEM)

Resin-cast etching was conducted in a similar fashion to Cui et al., 2022 [40] where 85% phosphoric acid (Sigma-Aldrich, San Jose, CA, USA) was diluted to a concentration of 37% using deionized water, and 4–5 mL was pipetted on top of the embedded specimens (one each of ORQ, ORQ-D, ORQ-TERI and ORQ-TERI-D) in a small beaker. After ten seconds, etching was stopped by adding 20 mL of deionized water to the beaker. Each sample was then removed from the bath and thoroughly rinsed with deionized water again. Samples were transferred to a pre-prepared mixture of 5% sodium hypochlorite solution and immersed for five minutes before a final rinse with deionized water. Samples were then left to dry in air for at least 12 h before mounting to an aluminum scanning electron microscopy stub using nickel paint and aluminum tape. Scanning electron microscopy was conducted in a JEOL 6610LV microscope using an accelerating voltage of 10 kV and working distance of 10 mm after sputtering with 10 nm of carbon.

### 2.8. Statistical Analysis

For statistical analysis, Sigma Plot 12.3 software (Sigma Plot Software, San Jose, CA, USA) was used. The analysis of homoscedasticity was performed using the Shapiro-Wilk test to distinguish between parametric and non-parametric data. For analysis of micro-CT parametric data of bone volume (BV), percentage of bone volume (BV/TV), trabecular bone thickness (Tb.Th), separation and number of trabeculae (TB.Sp and Tb.N) and least, total porosity (Po-tot), one-way ANOVA test and Tukey post-test were used. A significance level of *p* < 0.05 was adopted.

## 3. Results

### 3.1. Micro CT

Quantitatively, the mean bone volume (BV) obtained in the ORQ group was 0.03809 mm³, in the ORQ+D group was 0.04075 mm³, ORQTERI+D group with 0.04416 mm³ and a considerable increase in the ORQTERI group with 0.04959 mm³.

Presenting a statistically significant difference when comparing the ORQ and ORQTERI, ORQ and ORQTERI+D, ORQ+D and ORQTERI+D group (*p* < 0.05; Tukey post-test), (Figure 4).

This result pattern remained similar with the percentage of bone volume (BV/TV), with the averages of the ORQ, ORQ+D, ORQTERI and ORQTERI+D groups of 36.44%, 38.97%, 42.23% and 47.43 %, respectively, being the highest percetage of bone volume value obtained for the ORQTERI+D group. Having statistically significant difference when comparing the ORQ and ORQTERI, ORQ and ORQTERI+D, ORQ+D and ORQTERI+D group (*p* < 0.05; Tukey post-test), (Figure 5).

The mean value of trabecular bone thickness (Tb.Th) evaluated was 0.06103 mm for ORQ, 0.071 mm for ORQ+D, 0.09083 mm for ORQTERI and 0.09164 mm for ORQTERI+D. For such results, there was a statistically significant difference in the intragroup comparison between ORQ and ORQTERI, ORQ and ORTERI+D, ORQ+D and ORQ TERI, ORQ+D and ORQTERI+D (*p* < 0.05; Tukey post-test), (Figure 6).

For the results obtained regarding the number and separation of trabeculae (Tb.N and Tb.Sp), there was statistically significant difference between ORQ and ORQTERI, ORQ and ORQTERI+D (*p* < 0.05; Tukey post-test). In which the means obtained for the ORQ, ORQ+D, ORQTERI and ORQTERI groups +D in relation to Tb.N were 6.796 per mm, 5.966 per mm, 5.005 per mm and 5.53 per mm, and Tb.Sp were 0.1086 mm, 0.113 mm, 0.1238 mm and 0.1225 mm respectively (Figure 7 and Figure 8).

The average percentage of total porosity (Po.tot) in the peri-implant bone evaluated in the ORQ, ORQ+D, ORQTERI and ORQTERI+D groups was 65.94%, 61.02%, 57.76% and 52.56%, respectively. For these results, there was a statistically significant difference between the ORQ and ORQTERI, ORQ and ORQTERI+D, ORQ+D and ORQTERI+D groups (*p* < 0.05; Tukey’s post-test) (Figure 9).

Through three-dimensional reconstruction, it is also possible to visualize the representative image of the diferente groups: ORQ, ORQ+D, ORQTERI and ORQTERI+D (Figure 10).

### 3.2. Scanning Electron Microscopy (SEM)

Resin-cast etching of the peri-implant bone tissue revealed the underlying osteocyte networks in the newly formed bone. Secondary electron imaging of the bone-implant interface (Figure 11) showed osteocytes with a periodic spacing in the bone tissue and co-alignment with the implant surface, with the long axes of osteocytes running parallel to the implant (Figure 11A) in all treatment conditions. Osteocytes also partially bridged the gap between bone and implant with no intervening mineralized matrix (Figure 11B–D). Despite some preparation artifacts creating separation between bone and implant, evidence persisted of prior contact between the implant and either the body of the osteocyte or its cell processes. Local regions of both high and low canalicular density were also observed throughout the mineralized bone matrix.

Etching of the bone-implant interface showed the interaction between osteocytes and biological interfaces in the peri-implant environment. Figure 12A highlights one such region following Vitamin D treatment where bone apposition appears close to a large blood vessel. Osteocytes here reside in mineral-dense nodules, with a much higher canalicular density than the surrounding tissue. The bodies of osteocytes in this region resided on the edge of the hypermineralized nodule, located at the greatest possible distance from the blood vessel while remaining in the nodule. A closer view of the osteocyte-vessel interface (Figure 12B) showed evidence that cell processes permeate throughout the mineralized bone tissue and extend all the way into the blood vessel. Cell processes within the mineralized bone tissue formed a tortuous and twisting network through canaliculi in the tissue, with some distinct endpoints of cell processes visible at the surface of the etched tissue (Figure 12C).

## 4. Discussion

The morphometric evaluation showed an improvement in the effect of vitamin D supplementation in association with teriparatide, which resulted in better quality bone tissue around the implants installed in the systemic condition caused by orchiectomy. The parameters considered important to determine the quality of bone tissue according to Bouxsein et al., in 2010 [41] were selected, which were bone volume, percentage of bone volume, thickness of trabeculae, number and separation of trabeculae and percentage of total porosity.

The largest volume of bone tissue formed in the ORQTERI+D group, agreeing with the values for percentage of bone volume, and thus demonstrating a greater amount of bone formed for this group. However, the evaluation of the quality of the bone tissue must also be performed. For that, the parameters of trabecular thickness, number and spacing of trabeculae and the total porosity were evaluated. A bone with greater trabecular thickness, fewer total trabeculae, a smaller space between trabeculae, as well as a lower percentage of total porosity is characterized by having better quality [42]. This pattern of bone tissue was found in the ORQTERI+D group, followed by the ORQ+D and ORQTERI groups which, even with better results, approached the ORQ group. The last one presented the worst results regarding bone quality among the evaluated groups.

As expected, the presence of vitamin D and its action on the increase in the intestinal absorption of calcium supported the bone repair process around the installed implants, which can be noticed in all parameters evaluated, but it did not present a statistical difference when compared to the ORQ group. Therefore, the results showed the reduced bone quality pattern of the ORQ group, with greater porosity and thinner trabeculae with less volume.

The association of teriparatide with vitamin D shows, in fact, to potentiate its effects. In the result pattern of the ORQTERI+D group, in addition to the indirect effect of calcium uptake that occurs with vitamin D supplementation, the anabolic effects arising from teriparatide increase bone apposition, tubular reabsorption of calcium in the kidneys, and intestinal absorption of calcium—all of fundamental importance to obtain these results [30,31,34].

In this context, cases of osteoporosis or osteopenia, which present a decrease in bone mass and density, can cause a decrease in primary stability at the time of installation of osseointegrated implants in the jaws and, therefore, can increase their failure rate [1,42,43,44]. Drug therapies such as the combination of vitamin D and teriparatide, which increase the quantity and quality of bone tissue, should be further studied for evaluation in humans to increase the stability, support, and predictability of the procedure.

Clinical studies realized in male patients with established osteoporosis evaluated the effect of anabolic and anti-resorptive medications, such as teriparatide, alendronate sodium and a combination of these two, in the prevention of vertebral and long bone fractures. It was concluded that teriparatide administered alone is more effective in the treatment of osteoporosis, since it presented higher values of bone mineral density. The simultaneous use of a bisphosphonate and teriparatide presented worse results, as alendronate seems to neutralize the anabolic action of teriparatide on bone tissue [45,46].

Osteoporosis is still the subject of much discussion in the literature regarding rehabilitative treatment with osseointegrated implants [47,48,49]. Analyzing the percentage of contact between bone tissue and implants installed in the tibia of osteoporotic Wistar rats, reports have observed that the reduction in bone mass in osteoporotic rats promoted a smaller area of contact between the bone and the implant, which may cause less stability in the support of the prosthesis [2].

In addition, it was observed that in osteoporotic rats, treatment with alendronate sodium was not able to reverse osteoporosis, in which peri-implant bone repair in the tibia was delayed and with characteristics very similar to osteoporotic rats without drug treatment. As for the rats treated with raloxifene hydrochloride, both the peri-implant bone cytoarchitecture, as well as the histometric and immunohistochemical results, this medication outperformed the other groups (alendronate-treated and untreated osteoporotic rats) and resembled healthy rats (SHAM, control group) [49].

As the main cause of osteoporosis in women is the increase in bone resorption caused by menopause, in which trabecular bone is reduced by 20 to 30% and cortical bone by 5 to 10% [30], treatment should be with anti-resorptive drugs, such as mentioned above, through the administration of alendronate or raloxifene [5,49,50].

In men, bone loss is not critical when compared to women, therefore, osteopenia is caused by reduced bone formation [30], which justifies the study of anabolic drugs, that is, drugs that stimulate bone formation, such as teriparatide [30,31,50]. For both genders, when there is vitamin D deficiency, it must be supplemented to reestablish endogenous homeostasis, indirectly helping the bone formation process by increasing the serum calcium level [23,27].

Resin-cast etching of the bone-implant interface has previously shown the periodic arrangement and co-alignment of osteocytes as mechanosensory cells to drive bone formation and remodeling [51]. The etching technique has also been applied to investigate the role of matrix proteins on osteocyte morphology [52], osteocyte-to-osteocyte dendricity [53] and direct attachment to the implant interface [54]. In the resin-cast etched micrographs, dendritic extensions expand outwards from the body of the cell in a tortuous network throughout the mineralized bone matrix, possibly as a mechanism for cellular signalling.

In general, the efficacy of certain systemic treatment strategies for osteoporosis can rely on delivery of these molecules throughout the bone tissue. At larger length scales, blood vessels facilitate systemic delivery to the implant environment where the lacuno-canalicular and osteocyte networks form a finer pathway for molecular transport in the mineralized bone tissue. Resin-cast etching of peri-implant bone near these vascular features show casts of osteocyte processes extending into vasculature to link these two multiscale entities. While osteocytes have been characterized at the implant interface previously, observation of osteocyte interactions with peri-implant vasculature offers insight into the transport of Vitamin D, teriparatide, or other biomolecules from the vascular network into the local canalicular network to maintain tissue development at the implant interface. Where the body of most osteocytes in the resin-cast images reside tens of microns away from the blood vessel, this transport through the tissue may be required to influence osteocyte response at the cellular level.

The gold standard for evaluation of osseointegration is through histological analysis by light microscopy, where non-decalcified portions are used to measure the bone-implant interface. Scanning electron microscopy is also used to evaluate the bone healing process at the bone-implant interface in vivo. Transmission electron microscopy is the best method to evaluate the cells around the implant but can only visualize the interface without the implant being present and still only allows within a small area. In addition, resin cast etching is an effective technique that use acid etching to investigate 3D ultrastructure of osteocytes cells and their lacunar-canalicular [55].

The presence of vitamin D shows through acid-etching, osteocytes with higher canalicular density and interaction between osteocytes and biological interfaces in the peri-implant environment.

Therapy with supplementation of vitamin D and teriparatide together is an alternative that should be considered and studied further, for cases that require implant-supported oral rehabilitation in male patients with osteoporosis. Therefore, the encouraging results of this work should be considered to carry out new studies with a clinical focus, evaluating the feasibility of associating these drugs for treatments before and during therapy with dental implants in patients.

## 5. Conclusions

Through this in vivo study, it was possible to conclude that the treatment with vitamin D associated with teriparatide in rats with induced osteoporosis increases the volume and improves the bone quality around the implants installed in tibias.

## Figures and Tables

**Figure 1 biology-12-00228-f001:**
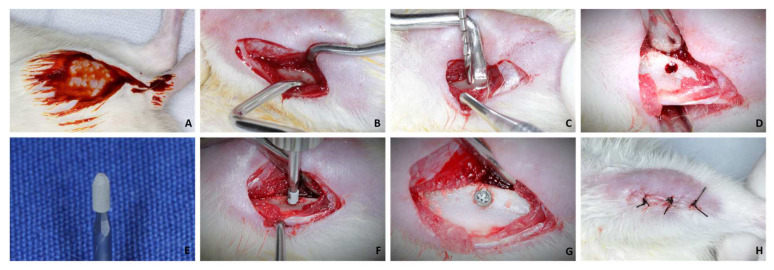
(**A**): Antisepsis with PVPI in both tibias after shaving; (**B**): Access to tibial metaphysis; (**C**,**D**): Milling with a 1.3 mm diameter drill and bed; (**E**): 1.5 × 3.5 mm implant; (**F**): Installation of the implant; (**G**): Implant installed at bone level; (**H**): Suture.

**Figure 2 biology-12-00228-f002:**
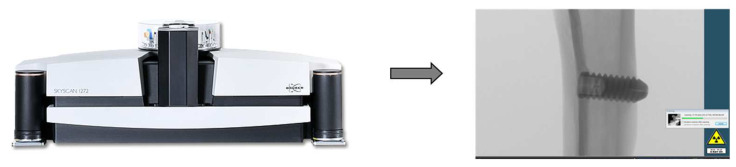
Scanning of parts through the SkyScan device and software.

**Figure 3 biology-12-00228-f003:**
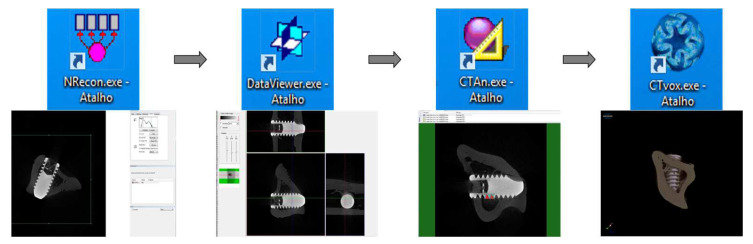
Sequence of software used for image reconstruction and analysis.

**Figure 4 biology-12-00228-f004:**
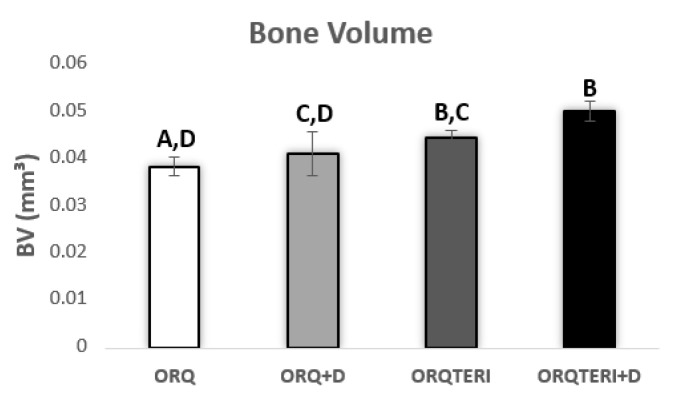
Bone volume graph. ORQ × ORQ+D × ORQTERI × ORQTERI+D. Statistical difference indicated by the letters (A, B, C and D).

**Figure 5 biology-12-00228-f005:**
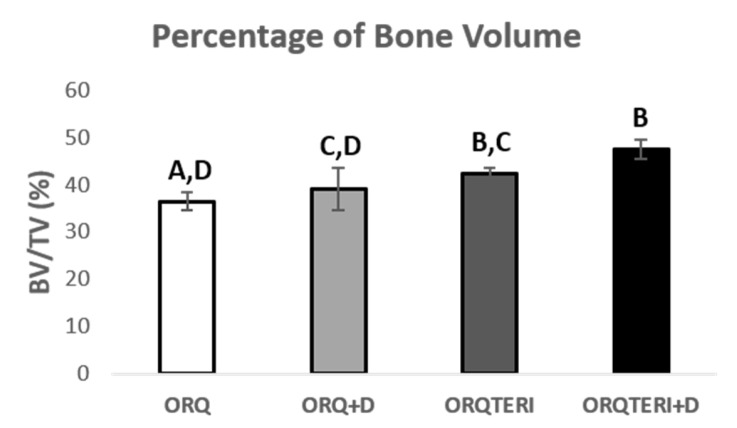
Percentage of bone volume. ORQ × ORQ+D × ORQTERI × ORQTERI+D. Statistical difference indicated by the letters (A, B, C and D).

**Figure 6 biology-12-00228-f006:**
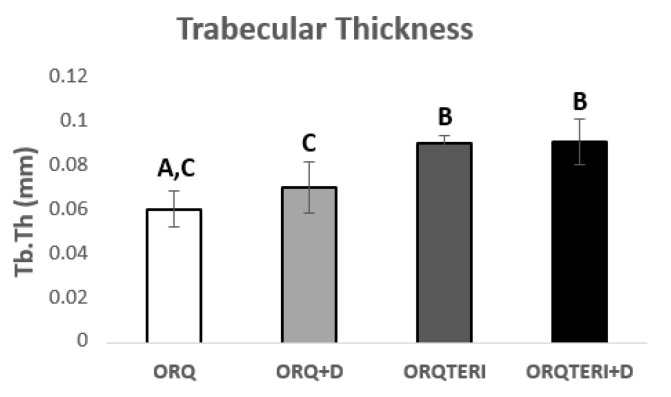
Comparison of the thickness between trabeculae. ORQ × ORQ+D × ORQTERI × ORQTERI+D. Statistical difference indicated by the letters (A, B and C).

**Figure 7 biology-12-00228-f007:**
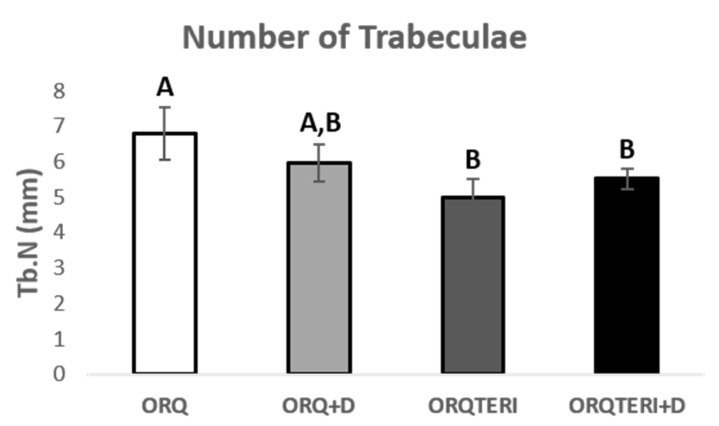
Number of trabeculae. ORQ × ORQ+D × ORQTERI × ORQTERI+D. Statistical difference indicated by the letters (A and B).

**Figure 8 biology-12-00228-f008:**
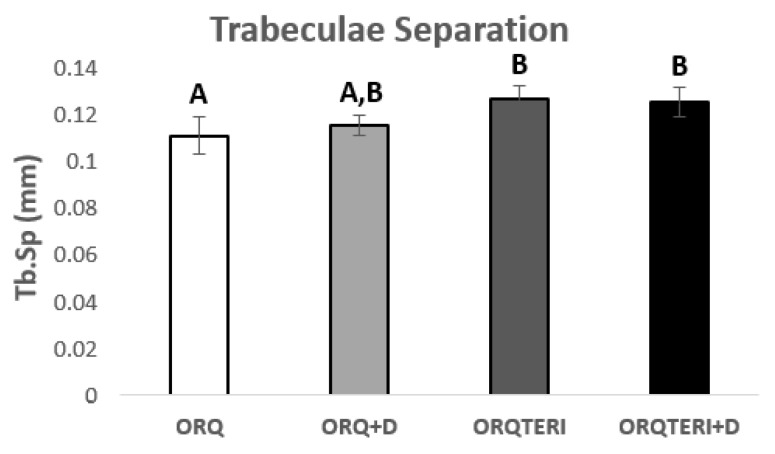
Comparison between the different results of trabecular separation. ORQ × ORQ+D × ORQTERI x ORQTERI+D. Statistical difference indicated by the letters (A and B).

**Figure 9 biology-12-00228-f009:**
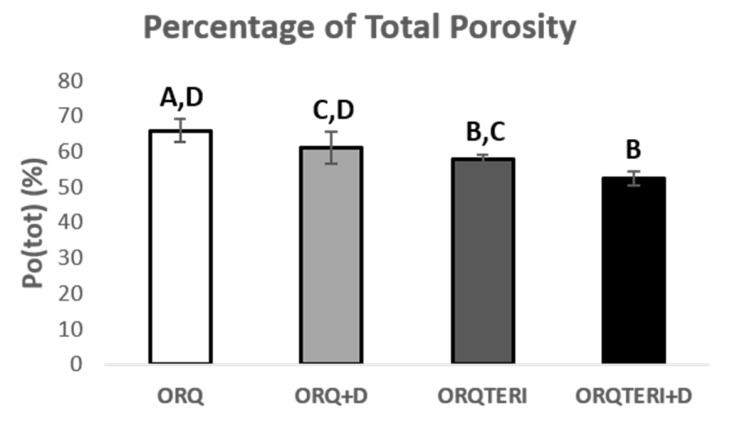
Percentage of total porosity. ORQ × ORQ+D × ORQTERI × ORQTERI+D. Statistical difference indicated by the letters (A, B, C and D).

**Figure 10 biology-12-00228-f010:**
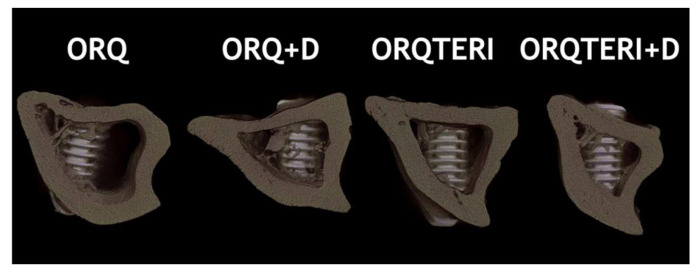
Image of three-dimensional reconstruction representing the differ- ent groups evaluated.

**Figure 11 biology-12-00228-f011:**
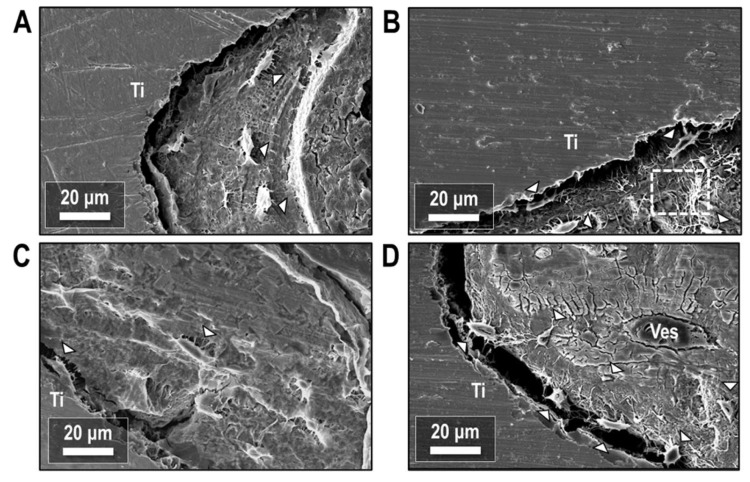
Secondary electron images of the bone-implant interface after resin cast etching. (**A**) ORQ, (**B**) ORQ-D, (**C**) ORQ-TERI and (**D**) ORQ-TERI-D. Ti: titanium; Ves: vessel. Osteocytes (white arrowheads) are periodically spaced in the bone matrix or bridge the gap between bone and implant. Adjacent regions of high and low canalicular density (box) are present within the mineralized tissue.

**Figure 12 biology-12-00228-f012:**
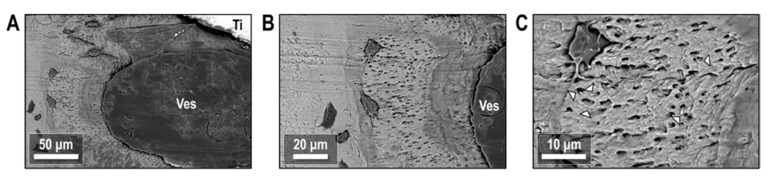
Backscattered electron images of peri-implant ORQ-D bone after resin cast etching. (**A**) Osteocytes resided in nodules of bone tissue with elevated mineral content adjacent to the implant and a blood vessel. (**B**) Canaliculi radiate outward from these osteocytes and permeate into the blood vessel. (**C**) Cell processes inside canaliculi form a tortuous network within the newly formed nodules, with distinct endpoints denoted with arrowheads. Ti: titanium; Ves: vessel.

**Table 1 biology-12-00228-t001:** Experimental groups according to treatment with vitamin D and/or teripatide.

Groups	Description
ORQ (n = 6)	Rats submitted to orchiectomy that did not receive drug treatment.
ORQ+D (n = 6)	Rats submitted to orchiectomy that received vitamin D (dose 0.1 mcg/kg/day).
ORQTERI (n = 6)	Rats submitted to orchiectomy that received teriparatide (dose 0.5 mcg/kg/day).
ORQTERI+D (n = 6)	Rats submitted to orchiectomy that received teriparatide (dose 0.5 mcg/kg/day) and vitamin D (dose 0.1 mcg/kg/day).

## Data Availability

The data presented in this study are available in this present paper.

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
