# Peer review of "Evaluation of Vitamin D isolated or Associated with Teriparatide in Peri-Implant Bone Repair in Tibia of Orchiectomized Rats"

_biology, 2023, doi:10.3390/biology12020228_

Round 1
Reviewer 1 Report
The article “Evaluation of vitamin D isolated or associated with teriparatide in peri-implant bone repair in tibia of orchiectomized rats” presented to me for evaluation is an extremely interesting, in its assumption, study. However, it requires a thorough rethink in terms of the form of its presentation. In its present form, with great caution, I recommend the manuscript for publication as an after major revision, and await the authors' response.
Details:
I consider the photographic documentation of the orchidectomy procedure to be infantile and completely unnecessary. Similarly, a description of the orchidectomy procedure is not necessary. It is a simple surgical procedure, widely described in veterinary textbooks.
In line 111, the authors inform about the 30-day period after the surgical procedures, after which the actual part of the experience related to experimental therapy begins. Explain what this time is for. If, in the authors' opinion, this was the time necessary to induce atrophic changes in the bone tissue after orchidectomy, then in my opinion it is absolutely too short.
What is the dose of Teriparatide used. Why 0.5 mcg/kg/day. The daily dose of Teriparatide in humans is 20 mcg. Assuming a human body weight of about 70 kg, based on the assumptions of the reviewed manuscript, the daily dose for humans would be 35 mcg, which is about 75% higher than the recommended one.
Why did the experimental therapy last 30 days?
Line 124 "After 30 days of drug treatment, the 24 rats underwent implant placement in the tibias" and next sentens "Eighteen commercially pure grade IV titanium implants were installed". What happened to the 6 rats?
Line 139. "Each animal received an implant in the tibial metaphysis" I'm guessing proximal, but how far from the state surface. How the reproducibility of the implant location in individual animals was determined. A 3.5 mm implant is a large object in terms of the size of a rat's bone. What was its impact on the mobility of the knee joint, and thus on the motility of the animal.
Line352 "However, in males, osteopenia is caused by reduced bone formation." There is a fundamental difference in the course of atrophic changes in the bone tissue of females and males in the conditions of afunction of the gonads. However, with such a thesis, it is wrong by definition. In males, the mechanism of osteoporosis also results from atrophic changes, although their course is different.
Line 397. "in rats with induced osteoporosis" In the materials and methods part, the authors inform that the experimental therapy was started 30 days after the orchidectomy. This period is too short to cause osteoporosis. In addition, the sham-operated group was not planned in the experiment, which could be the basis for determining the existence of atrophic changes of an osteoporotic nature. I think the lack of this group is a big mistake. Therefore, the firm statement “in rats with induced osteoporosis” is unjustified.
Author Response
Reviewer #1:
Comments for the author:
1-Reviewer: The article “Evaluation of vitamin D isolated or associated with teriparatide in peri-implant bone repair in tibia of orchiectomized rats” presented to me for evaluation is an extremely interesting, in its assumption, study. However, it requires a thorough rethink in terms of the form of its presentation. In its present form, with great caution, I recommend the manuscript for publication as an after major revision, and await the authors' response.
Details:
I consider the photographic documentation of the orchidectomy procedure to be infantile and completely unnecessary. Similarly, a description of the orchidectomy procedure is not necessary. It is a simple surgical procedure, widely described in veterinary textbooks.
Response: Dear reviewer, as requested, the image of the orchiectomy was removed, but the context of what was done, that is, the procedure was maintained in order to elucidate all the steps taken in the work.
2-Reviewer: In line 111, the authors inform about the 30-day period after the surgical procedures, after which the actual part of the experience related to experimental therapy begins. Explain what this time is for. If, in the authors' opinion, this was the time necessary to induce atrophic changes in the bone tissue after orchidectomy, then in my opinion it is absolutely too short.
Response: Dear reviewer, the experimental design proposed in this study is the model that we have used in our lab and we have previous studies published of our group1-3. The period of 30 days after orchiectomy promotes changes in bone repair after tooth extraction and after implant's installation. That is because we kept the same design in the present study.
- de Oliveira Puttini I, Gomes-Ferreira PHDS, de Oliveira D, Hassumi JS, Gonçalves PZ, Okamoto R. Teriparatide improves alveolar bone modelling after tooth extraction in orchiectomized rats. Arch Oral Biol. 2019 Jun;102:147-154. doi: 10.1016/j.archoralbio.2019.04.007. Epub 2019 Apr 16. PMID: 31022626.
- de Oliveira D, de Oliveira Puttini I, Silva Gomes-Ferreira PH, Palin LP, Matsumoto MA, Okamoto R. Effect of intermittent teriparatide (PTH 1-34) on the alveolar healing process in orchiectomized rats. Clin Oral Investig. 2019 May;23(5):2313-2322. doi: 10.1007/s00784-018-2672-y. Epub 2018 Oct 6. PMID: 30291494.
- Gomes-Ferreira, P.H.S.; de Oliveira, D.; Frigério, P.B.; de Souza Batista. R.; Grandfield, K.; Okamoto, R. Teriparatide improves microarchitectural characteristics of peri-implant bone in orchiectomized rats. Osteoporos Int 2020 31(9):1807-1815.
3-Reviewer: What is the dose of Teriparatide used. Why 0.5 mcg/kg/day. The daily dose of Teriparatide in humans is 20 mcg. Assuming a human body weight of about 70 kg, based on the assumptions of the reviewed manuscript, the daily dose for humans would be 35 mcg, which is about 75% higher than the recommended one. Why did the experimental therapy last 30 days?
Response: The dose for application in rats was 60µg/kg, and the dose for humans in a usual treatment with PTH 1-34 is 20-40µg daily. The rats in the present study weighed about 500g or more, that is, these animals would receive more than the usual minimum dose for humans (about 30µg/day). With this in mind and the metabolism of the rat, together with a veterinarian, the dose of the present study was defined, which has already been described in the literature in two studies that evaluated the repair of post-exodontic alveoli and dental implants in tibial metaphysis (mentioned below).
de Oliveira Puttini I, Gomes-Ferreira PHDS, de Oliveira D, Hassumi JS, Gonçalves PZ, Okamoto R. Teriparatide improves alveolar bone modelling after tooth extraction in orchiectomized rats. Arch Oral Biol. 2019 Jun;102:147-154. doi: 10.1016/j.archoralbio.2019.04.007. Epub 2019 Apr 16. PMID: 31022626.
de Oliveira D, de Oliveira Puttini I, Silva Gomes-Ferreira PH, Palin LP, Matsumoto MA, Okamoto R. Effect of intermittent teriparatide (PTH 1-34) on the alveolar healing process in orchiectomized rats. Clin Oral Investig. 2019 May;23(5):2313-2322. doi: 10.1007/s00784-018-2672-y. Epub 2018 Oct 6. PMID: 30291494.
Gomes-Ferreira, P.H.S.; de Oliveira, D.; Frigério, P.B.; de Souza Batista. F.R.; Grandfield, K.; Okamoto, R. Teriparatide improves microarchitectural characteristics of peri-implant bone in orchiectomized rats. Osteoporos Int 2020 31(9):1807-1815.
4-Reviewer: Line 124 "After 30 days of drug treatment, the 24 rats underwent implant placement in the tibias" and next sentens "Eighteen commercially pure grade IV titanium implants were installed". What happened to the 6 rats?
Response: There was an error in the number of implants, we thank you for your keen assessment and help in making an article with minimal errors possible for publication. The sentence was corrected by changing 18 to 24 implants
5-Reviewer: Line 139. "Each animal received an implant in the tibial metaphysis" I'm guessing proximal, but how far from the state surface. How the reproducibility of the implant location in individual animals was determined. A 3.5 mm implant is a large object in terms of the size of a rat's bone. What was its impact on the mobility of the knee joint, and thus on the motility of the animal.
Response: Dear reviewer, it is a very plausible concern. We have extensive experience in the surgical procedure for installation in the tibial metaphysis of rats, and this is an important point to evaluate in the postoperative period. We usually install the implants in the proximal tibial metaphysis, 0.5 cm below the capsule. In the postoperative period, the animals have their mobility preserved after recovery from the surgical wound.
6-Reviewer: Line352 "However, in males, osteopenia is caused by reduced bone formation." There is a fundamental difference in the course of atrophic changes in the bone tissue of females and males in the conditions of afunction of the gonads. However, with such a thesis, it is wrong by definition. In males, the mechanism of osteoporosis also results from atrophic changes, although their course is different.
Response: Thank you for your comment, a better explanation has been added to the text regarding what the authors wanted to cite. Farahmand, et al., 2016 mentions that women are the most affected by bone resorption, resulting from menopause, while men's bone loss is less dramatic, being initially affected by the decline in bone formation, thus justifying the use of anabolic bone for the treatment, where the main goal is to form new bone. You can evaluate is was changed the sentence in line 352 – 357.
“As the main cause of osteoporosis in women is the increase in bone resorption caused by menopause, in which trabecular bone is reduced by 20 to 30% and cortical bone by 5 to 10% [30], treatment should be with anti-resorptive drugs, such as mentioned above, through the administration of alendronate or raloxifene [5,49,50].
In men, bone loss is not critical when compared to women, therefore, osteopenia is caused by reduced bone formation [30], which justifies the study of anabolic drugs, that is, drugs that stimulate bone formation, such as teriparatide [30,31,50]. For both genders, when there is vitamin D deficiency, it must be supplemented to reestablish endogenous homeostasis, indirectly helping the bone formation process by increasing the serum calcium level [23,27]. “
7-Reviewer: Line 397. "in rats with induced osteoporosis" In the materials and methods part, the authors inform that the experimental therapy was started 30 days after the orchidectomy. This period is too short to cause osteoporosis. In addition, the sham-operated group was not planned in the experiment, which could be the basis for determining the existence of atrophic changes of an osteoporotic nature. I think the lack of this group is a big mistake. Therefore, the firm statement “in rats with induced osteoporosis” is unjustified.
Response: We agree that the period used in this study and in other studies from our group is capable to cause an osteopenic condition that evokes an impairment of bone repair. This condition, if maintained for longer periods, will lead to osteoporosis condition. But as our focus is on bone repair, we consider that the present experimental design contributes to evaluate this condition.

Reviewer 2 Report
This article, entitled "Evaluation of vitamin D isolated or associated with teriparatide in peri-implant bone repair in tibia of orchiectomized rats" provides an experimental study on rats of the quality of bone tissue of peri-implant bone in orchiectomized rats, treated with vitamin D isolated or associated with teriparatide, an anabolic agent that promotes bone formation directly by activating osteoblasts and also indirectly by increasing calcium levels due to an effect on both renal tubular reabsorption and intestinal absorption. The authors realized a comparative analysis of four groups of rats: 1) orchiectomized without treatment, 2) orchiectomized treated with vitamin D, 3) orchiectomized treated with teriparatide and 4) orchiectomized treated with both teriparatide and vitamin D. Bone samples were analyzed by micro-CT and data obtained were analyzed by 1-way ANOVA and Tukey post-test, with a significance level of 5%. The results obtained indicated that treatment with vitamin D associated with teriparatide increases bone volume and improves bone quality. Overall, the paper is interesting from a medical point of view since and the manuscript contains sufficient noteworthy information to justify publication. The subject is significant and concisely stated. I have only minor points to improve the manuscript:
1. The statistical analysis used, 1-way ANOVA and Tukey post-test, is correct but I think that it can be much improved by the use a two-way ANOVA because this analysis allows to assess the main effect of each treatment and also if there is any interaction, positive or negative, between them. , and this is a fact of main importance.
2. Figure 1 of bilateral orchiectomy is not necessary since bilateral orchiectomy is a very common technique widely used
3. A figure with one representative image obtained by micro-CT of each experimental group should be added
Author Response
Reviewer #2:
Comments for the author:
1-Reviewer: This article, entitled "Evaluation of vitamin D isolated or associated with teriparatide in peri-implant bone repair in tibia of orchiectomized rats" provides an experimental study on rats of the quality of bone tissue of peri-implant bone in orchiectomized rats, treated with vitamin D isolated or associated with teriparatide, an anabolic agent that promotes bone formation directly by activating osteoblasts and also indirectly by increasing calcium levels due to an effect on both renal tubular reabsorption and intestinal absorption. The authors realized a comparative analysis of four groups of rats: 1) orchiectomized without treatment, 2) orchiectomized treated with vitamin D, 3) orchiectomized treated with teriparatide and 4) orchiectomized treated with both teriparatide and vitamin D. Bone samples were analyzed by micro-CT and data obtained were analyzed by 1-way ANOVA and Tukey post-test, with a significance level of 5%. The results obtained indicated that treatment with vitamin D associated with teriparatide increases bone volume and improves bone quality. Overall, the paper is interesting from a medical point of view since and the manuscript contains sufficient noteworthy information to justify publication. The subject is significant and concisely stated. I have only minor points to improve the manuscript:
- The statistical analysis used, 1-way ANOVA and Tukey post-test, is correct but I think that it can be much improved by the use a two-way ANOVA because this analysis allows to assess the main effect of each treatment and also if there is any interaction, positive or negative, between them. , and this is a fact of main importance.
Response: We would like to thank you for all the revisions expressed, we are very happy to be able to improve this paper more and more in order to have a better publication. Regarding the two-way ANOVA test, we liked the suggestion and we did the test, after the final result we obtained very similar results to the one-way ANOVA, we believe that this is due to the presence of only one variant which is the drug treatment, if we had another variant such as surface treatment or euthanasia times, we believe that the difference would be more expressive. For this reason, I attached the analysis carried out as a supplementary document so that the present reviewer can evaluate it, but we will choose to keep the results as they are. Despite this decision we believe it was an excellent suggestion, thank you very much.
2-Reviewer: Figure 1 of bilateral orchiectomy is not necessary since bilateral orchiectomy is a very common technique widely used
Response: Dear reviewer, as requested, the image of the orchiectomy was removed,
3-Reviewer: A figure with one representative image obtained by micro-CT of each experimental group should be added
Response: As requested the representative image of the three-dimensional microtomography was added after the results of Micro-CT.

Round 2
Reviewer 1 Report
No suggestions